# Full-length genome characterization and phylogenetic analysis of SARS-CoV-2 virus strains from Yogyakarta and Central Java, Indonesia

Gunadi[1], Hendra Wibawa[2], Marcellus[1], Mohamad Saifudin Hakim[3], Edwin Widyanto Daniwijaya[4], Ludhang Pradipta Rizki[3], Endah Supriyati[5], Dwi Aris Agung Nugrahaningsih[6], Afiahayati[7], Siswanto[8], Kristy Iskandar[9], Nungki Anggorowati[10], Alvin Santoso Kalim[1], Dyah Ayu Puspitarani[1], Kemala Athollah[1], Eggi Arguni[11], Titik Nuryastuti[3] and Tri Wibawa[3]

[1] Pediatric Surgery Division, Department of Surgery, Faculty of Medicine, Public Health and Nursing, Universitas Gadjah Mada, Yogyakarta, Indonesia
[2] Disease Investigation Center Wates, Yogyakarta, Ministry of Agriculture, Indonesia
[3] Department of Microbiology, Faculty of Medicine, Public Health and Nursing, Universitas Gadjah Mada, Yogyakarta, Indonesia
[4] Department of Microbiology, Faculty of Medicine, Public Health and Nursing, Universitas Gadjah Mada/UGM Academic Hospital, Yogyakarta, Indonesia
[5] Centre of Tropical Medicine, Faculty of Medicine, Public Health and Nursing, Universitas Gadjah Mada, Yogyakarta, Indonesia
[6] Department of Pharmacology and Therapy, Faculty of Medicine, Public Health and Nursing, Universitas Gadjah Mada, Yogyakarta, Indonesia
[7] Department of Computer Science and Electronics, Faculty of Mathematics and Natural Sciences, Universitas Gadjah Mada, Yogyakarta, Indonesia
[8] Department of Physiology, Faculty of Medicine, Public Health and Nursing, Universitas Gadjah Mada/UGM Academic Hospital, Yogyakarta, Indonesia
[9] Department of Child Health, Faculty of Medicine, Public Health and Nursing, Universitas Gadjah Mada/UGM Academic Hospital, Yogyakarta, Indonesia
[10] Department of Anatomical Pathology, Faculty of Medicine, Public Health and Nursing, Universitas Gadjah Mada, Yogyakarta, Indonesia
[11] Department of Child Health, Faculty of Medicine, Public Health and Nursing, Universitas Gadjah Mada, Yogyakarta, Indonesia

Corresponding authors
Gunadi, drgunadi@ugm.ac.id
Hendra Wibawa, hendra.wibawa@pertanian.go.id

## ABSTRACT

**Background:** Recently, SARS-CoV-2 virus with the D614G mutation has become a public concern due to rapid dissemination of this variant across many countries. Our study aims were (1) to report full-length genome sequences of SARS-CoV-2 collected from four COVID-19 patients in the Special Region of Yogyakarta and Central Java provinces, Indonesia; (2) to compare the clade distribution of full-length genome sequences from Indonesia (*n* = 60) from March to September 2020 and (3) to perform phylogenetic analysis of SARS-CoV-2 complete genomes from different countries, including Indonesia.

**Methods:** Whole genome sequencing (WGS) was performed using next-generation sequencing (NGS) applied in the Illumina MiSeq instrument. Full-length virus genomes were annotated using the reference genome of hCoV-19/Wuhan/Hu-1/2019 (NC_045512.2) and then visualized in UGENE v. 1.30. For phylogenetic

analysis, a dataset of 88 available SARS-CoV-2 complete genomes from different countries, including Indonesia, was retrieved from GISAID.
**Results:** All patients were hospitalized with various severities of COVID-19. Phylogenetic analysis revealed that one and three virus samples belong to clade L and GH. These three clade GH virus samples (EPI_ISL_525492, EPI_ISL_516800 and EPI_ISL_516829) were not only located in a cluster with SARS-CoV-2 genomes from Asia but also those from Europe, whereas the clade L virus sample (EPI_ISL_516806) was located amongst SARS-CoV-2 genomes from Asia. Using full-length sequences available in the GISAID EpiCoV Database, 39 of 60 SARS-CoV-2 (65%) from Indonesia harbor the D614G mutation.
**Conclusion:** These findings indicate that SARS-CoV-2 with the D614G mutation appears to become the major circulating virus in Indonesia, concurrent with the COVID-19 situation worldwide.

## INTRODUCTION

In December 2019, an outbreak of Severe Acute Respiratory Syndrome Coronavirus 2 (SARS-CoV-2) causing Coronavirus Disease 2019 (COVID-19) was detected in Wuhan, China and has become a global pandemic, including Indonesia (*World Health Organization, 2020a*; *Phelan, Katz & Gostin, 2020*).

In Indonesia, the first two COVID-19 cases were reported on 2 March 2020. Since then, the confirmed cases have been continuously increasing although several public health measures, involving isolation of confirmed patients and community wide containment in addition to strictly enforced personal health protocols, were conducted to halt transmission events (*World Health Organization, 2020b*). Tragically, on 19 November 2020, Indonesia recorded 478,720 COVID-19 infections and 15,503 deaths (*World Health Organization, 2020b*). This situation means that Indonesia has reported the second most confirmed COVID-19 cases in the South East Asia countries after the Philippines, yet has the highest number of deaths caused by COVID-19 among other South East Asia countries (*World Health Organization, 2020b*).

Just as many other countries, the detection of SARS-CoV-2 in suspected people is mainly based on reverse transcriptase real-time polymerase chain reaction (RT-PCR). The supply of PCR reagents, trained lab personnel and the availability of laboratories with sufficient biocontainment levels are major challenges of SARS-CoV-2 detection in developing countries, such as Indonesia (*Younes et al., 2020*). Therefore, it is not surprising that the tested people per week is still lower than the World Health Organization (WHO) standard (*World Health Organization, 2020b*).

Recently, SARS-CoV-2 with the D614G mutation became the most frequently detected globally, including South East Asia region (*Korber et al., 2020*; *Nguyen et al., 2020*). Interestingly, SARS-CoV-2 with the G614 variant had significantly higher infectious titers

than the original D614 virus, and COVID-19 patients with the G614 variant had a higher viral load than patients without the mutation (*Korber et al., 2020*). A recent study showed that the SARS-CoV-2 with the G614 variant revealed increased infectivity, competitive fitness, and transmission than the wild-type D614 virus in human airway epithelial cells and hamster (*Hou et al., 2020*). However, this mutation was not associated with the severity of COVID-19 (*Korber et al., 2020*; *Nguyen et al., 2020*). Here, we aimed: (1) to report full-length genome sequences of SARS-CoV-2 collected from four COVID-19 patients in the Special Region of Yogyakarta and Central Java provinces, Indonesia; (2) to compare the clade distribution of full-length genome sequences from Indonesia ($n$ = 60) from March to September 2020; and (3) to perform phylogenetic analysis of SARS-CoV-2 complete genomes from different countries, including Indonesia.

# MATERIALS AND METHODS

## Classification of COVID-19 severity

We determined the COVID-19 severity according to the WHO classifications: (1) mild, symptomatic COVID-19 patients without evidence of hypoxia or pneumonia; (2) moderate, clinical signs of pneumonia (i.e., fever, cough, dyspnea, fast breathing) but not severe pneumonia, including blood oxygen saturation levels ($SpO_2$) ≥90% in room air; (3) severe, clinical signs of pneumonia plus one of the conditions as follows: respiratory rate >30 breaths/minute, severe respiratory distress, or $SpO_2$ <90% in room air and (4) critical, Acute Respiratory Distress Syndrome, sepsis, or septic shock, other complications such as acute pulmonary embolism, acute coronary syndrome, acute stroke and delirium (*Beeching, Fletcher & Fowler, 2019*).

## Virus samples

All virus samples were collected from hospitalized patients with COVID-19 from June to August 2020 in Yogyakarta and Central Java provinces. Samples were collected from nasopharyngeal swabs and then directly put into viral transport media (DNA/RNA Shield™ Collection Tube with Swab, Zymo Research, CA, USA). Samples were sent to the Department of Microbiology and Laboratorium Diagnostik Yayasan Tahija World Mosquito Program, Faculty of Medicine, Public Health and Nursing, Universitas Gadjah Mada and the Disease Investigation Center, Wates, Yogyakarta for SARS-CoV-2 virus detection using Real-Q 2019-nCoV Detection Kit (BioSewoom, Seoul, South Korea) with LightCycler® 480 Instrument II (Roche Diagnostics, Mannheim, Germany).

## Whole genome sequencing

Total viral RNA was extracted from 15 original samples (nasopharyngeal swabs) using a QiAMP Viral RNA mini kit (Qiagen, Hilden, Germany), followed by double stranded cDNA synthesis using Maxima H Minus Double-Stranded cDNA Synthesis (Thermo Fisher Scientific, MA, USA), and then purified by a GeneJET PCR Purification Kit (Thermo Fisher Scientific, MA, USA). The Nextera DNA Flex for Enrichment using Respiratory Virus Oligos Panel was used for library preparations, and whole genome sequencing was performed using next generation sequencing (NGS) applied in the

Illumina MiSeq instrument (Illumina, San Diego, CA, USA) with Illumina MiSeq reagents v3 150 cycles (2 × 75 cycles). Among 15 samples that were analyzed by NGS, only four samples showed good data for further bioinformatics analysis. The paired reads were trimmed for quality and length and assembled by mapping to the reference genome from Wuhan, China (hCoV-19/Wuhan/Hu-1/2019, GenBank accession number: NC_045512.2) using Burrow-Wheeler Aligner (BWA) algorithm embedded in UGENE v. 1.30 (*Unipro UGENE Online User Manual, 2020*). Single nucleotide polymorphism (SNP) was identified based on the number of high confidence base calls (consensus sequence variations of the assembly) that disagree with the reference bases for the genome position of interest. These variations were then exported to a vcf file and visualized in MS Excel. All four full-genome sequences of SARS-CoV-2 had the following accession IDs: EPI_ISL_516800, EPI_ISL_516806, EPI_ISL_516829 and EPI_ISL_525492 (*GISAID, 2020*).

### Genome annotation and phylogenetic analysis

Full-length virus genomes were annotated using the reference genome of hCoV-19/ Wuhan/Hu-1/2019 (NC_045512.2). For phylogenetic analysis, a dataset of 88 available SARS-CoV-2 virus genomes from different countries, including those from Indonesia, was retrieved from GISAID. Instead of using all available sequences in GISAID, we used sequences of several viruses representing SARS-CoV-2 clades from some countries that have complete genome, high-coverage, and no stretches of "NNNN" for the phylogenetic dataset. Sequence alignment was performed using the MAFFT program server for multiple nucleotide sequence alignment (https://mafft.cbrc.jp/alignment/server/). A phylogenetic tree was constructed using 29.400 nucleotide length starting from the ORF1ab open reading frame of SARS-CoV-2 using a maximum likelihood statistical method with 1,000 bootstrap replications and selected the best-fitting substitution model (GTR+G+I) for the dataset. All the analyses were performed in Molecular Evolutionary Genetics Analysis version 10 (MEGA X) software (*Kumar et al., 2018*). Because the purpose of this phylogenetic analysis was to determine the evolutionary relationships between our virus samples and the other SARS-CoV-2 viruses, the tree was rooted to the oldest virus, hCoV-19/Wuhan/Hu-1/2019.

### Ethical approval

The Medical and Health Research Ethics Committee of the Faculty of Medicine, Public Health and Nursing, Universitas Gadjah Mada/Dr. Sardjito Hospital approved this study (KE/FK/0563/EC/2020). Written informed consent was obtained from all participants before joining in this study.

## RESULTS

### Whole genome sequences of SARS-CoV-2 from the special region of Yogyakarta and Central Java provinces, Indonesia

All patients were classified as moderate COVID-19, except patient-4 as a mild case (Table 1; Table S1). The details of case presentations are described in the Table S1.

**Table 1 Characteristics of four patients with COVID-19 and SARS-CoV-2 virus samples from Yogyakarta and Central Java.**

| Patient No | Sex | Age (year) | COVID-19 severity | $C_T$ value | Virus name (GISAID Accession ID) | Average coverage | Collection date | Lineage (GISAID clade) | Amino acid mutations* (no. mutation and position of proteins-*encoded genes*) |
|---|---|---|---|---|---|---|---|---|---|
| 1 | Female | 83 | Moderate | 16.9 | hCoV19/Indonesia/ YO-UGM-781481/2020 (EPI_ISL_516829) | 3748x | 10 August 2020 | GH | 9: **NSP3**-*ORF1ab* (P679S), **NSP12**-*ORF1ab* (P323L, A656S), **NSP13**-*ORF1ab* (M576I), **Spike**-*S* (D614G), **NS3**-*ORF3a* (A54V, Q57H, A99S), **NP**-*N* (Q160R) |
| 2 | Male | 77 | Moderate | 19.7 | hCoV19/Indonesia/ YO-UGM-202449/2020 (EPI_ISL_516800) | 22088x | 22 June 2020 | GH | 4: **NSP3**-*ORF1ab* (P822L), **NSP12**-*ORF1ab* (P323L), **Spike**-*S* (D614G), **NS3**-*ORF3a* (Q57H) |
| 3 | Female | 55 | Moderate | 24.7 | hCov19/Indonesia/ JT-UGM-202538/ 2020 (EPI_ISL_525492) | 347x | 26 June 2020 | GH | 5: **NSP3**-*ORF1ab* (P822L), **NSP12**-*ORF1ab* (P323L), **Spike**-*S* (D614G), **NS3**-*ORF3a* (Q57H), **NS7a**-ORF7a (H73Y) |
| 4 | Male | 30 | Mild | 27.9 | hCoV19/Indonesia/ YO-UGM-200927/2020 (EPI_ISL_516806) | 102x | 16 May 2020 | L | 1: **NSP5**-*ORF1ab* (M49I) |

**Notes:**
* Name of SARS-CoV-2 proteins (bold) are followed by related genes (italic) and amino acid mutation indicated in bracket.
CT, cycle threshold.
**Ref. sequence: hCoV-19/Wuhan/Hu-1/2019 (NC_045512.2).**

WGS revealed that the virus sample collected from patient-1 (hCoV19/Indonesia/YO-UGM-781481/2020, ID: EPI_ISL_516829), patient-2 (hCoV19/Indonesia/YO-UGM-202449/2020, ID: EPI_ISL_516800) and patient-3 belonged to the GH clade, while those from patient-4 (hCoV19/Indonesia/YO-UGM-200927/2020, ID: EPI_ISL_516806) showed the L clade.

Moreover, WGS of virus from patient-1, patient-2 and patient 3 showed nine amino acid mutations in six proteins, including NSP3 (P679S), NSP12 (P323L, A656S), NSP13 (M576I), spike (D614G), NS3 (A54V, Q57H, A99S), and NP (Q160R); four amino acid mutations in four proteins: NSP3 (P822L), NSP12 (P323L), Spike (D614G) and NS3 (Q57H); and five amino acid mutations in five proteins: NSP3 (P822L), NSP12 (P323L), Spike (D614G), NS3 (Q57H) and NS7a (H73Y), respectively; whereas those from patient-4 consisted of only one mutation in the NSP5 protein (M49I) (Table 1).

The genome-wide SNPs and amino acid variations of our samples are shown in Tables 2 and 3, respectively (positions referred to the reference sequence: NC_045512.2). Not all SNPs cause amino acid changes in our samples.

## Clade distribution of full-length genome sequences from Indonesia

Whole genome sequencing revealed that one virus (hCoV19/Indonesia/YO-202449/2020, EPI_ISL_516800) had a complete SARS-CoV-2 genome (29.903 nt). Although the other three virus samples were shorter due to incomplete UTRs at either the 5′ or 3′, they possessed full-length and complete open reading frames (ORFs) with a size of 29.409 nt

**Table 2** The genome-wide SNPs of four SARS-CoV-2 virus samples from Yogyakarta and Central Java (positions referred to the reference sequence: NC_045512.2).

| No | Virus ID | 5′ UTR | | NSP3-*ORF1ab* | | | | NSP5-*ORF1ab* | | NSP12-*ORF1ab* | | | | | NSP13-*ORF1ab* | NSP14-*ORF1ab* | | Spike-*S* | NS3-*ORF3a* | | | Matrix-*M* | | NS7a-*ORF7a* | NP-*N* | |
|---|---|---|---|---|---|---|---|---|---|---|---|---|---|---|---|---|---|---|---|---|---|---|---|---|---|---|
| | | 26 | 241 | 3037 | 3529 | 4754 | 5184 | 10201 | 10507 | 14055 | 14292 | 14408 | 14694 | 15406 | 17964 | 18744 | 18877 | 23403 | 25553 | 25563 | 25687 | 26735 | 26867 | 27610 | 28735 | 28752 |
| 1 | hCoV-19/Wuhan/Hu-1/2019 (NC_045512.2) | A | C | C | T | C | C | G | C | G | C | C | C | G | G | C | C | A | C | G | G | C | A | C | T | A |
| 2 | hCoV19/Indonesia/YO-UGM-781481/2020 (EPI_ISL_516829) | A | **T** | **T** | **C** | **T** | C | G | C | G | **T** | **T** | **T** | **T** | **T** | C | **T** | **G** | **T** | **T** | **T** | **T** | A | C | **C** | **G** |
| 3 | hCoV19/Indonesia/YO-UGM-202449/2020 (EPI_ISL_516800) | A | **T** | **T** | T | C | **T** | G | **T** | G | C | **T** | C | G | G | **T** | **T** | **G** | C | **T** | G | **T** | **G** | C | T | A |
| 4 | hCov19/Indonesia/JT-UGM-202538/2020 (EPI_ISL_525492) | **G** | **T** | **T** | T | C | **T** | G | **T** | **T** | C | **T** | C | G | G | **T** | **T** | **G** | C | **T** | G | **T** | **G** | **T** | T | A |
| 5 | hCoV19/Indonesia/YO-UGM-200927/2020 (EPI_ISL_516806) | A | C | C | T | C | C | **T** | C | G | C | C | C | G | G | C | C | A | C | G | G | C | A | C | T | A |

**Notes:**
Name of SARS-CoV-2 proteins (bold) are followed by related genes (italic) and nucleotide variations are colour-shaded.
SNPs are shown according to nucleotide positions starting from 5′-UTR (untranslated region) of SARS-CoV-2 genome.

Table 3 **The amino acid variations of four SARS-CoV-2 virus samples from Yogyakarta and Central Java (positions referred to the reference sequence: NC_045512.2).**

| No | Virus ID | NSP3-*ORF1ab* | | | | NSP5-*ORF1ab* | | NSP12-*ORF1ab* | | | | | NSP13-*ORF1ab* | NSP14-*ORF1ab* | | Spike-*S* | NS3-*ORF3a* | | | Matrix-*M* | | NS7a-*ORF7a* | NP-*N* | |
|---|---|---|---|---|---|---|---|---|---|---|---|---|---|---|---|---|---|---|---|---|---|---|---|---|
| | | **106** | **270** | **679** | **822** | **49** | **151** | **205** | **284** | **323** | **418** | **656** | **576** | **235** | **280** | **614** | **54** | **57** | **99** | **71** | **115** | **73** | **154** | **160** |
| 1 | hCoV-19/Wuhan/Hu-1/2019 (NC_045512.2) | F | D | P | P | M | N | L | D | P | D | A | M | Y | L | D | A | Q | A | Y | E | H | N | Q |
| 2 | hCoV19/Indonesia/YO-UGM-781481/2020 (EPI_ISL_516829) | F | D | **S** | P | M | N | L | D | **L** | D | **S** | **I** | Y | L | **G** | **V** | **H** | **S** | Y | E | H | N | **R** |
| 3 | hCoV19/Indonesia/YO-UGM-202449/2020 (EPI_ISL_516800) | F | D | P | **L** | M | N | L | D | **L** | D | A | M | Y | L | **G** | A | **H** | A | Y | E | H | N | Q |
| 4 | hCov19/Indonesia/JT-UGM-202538/2020 (EPI_ISL_525492) | F | D | P | **L** | M | N | L | D | **L** | D | A | M | Y | L | **G** | A | **H** | A | Y | E | **Y** | N | Q |
| 5 | hCoV19/Indonesia/YO-UGM-200927/2020 (EPI_ISL_516806) | F | D | P | P | **I** | N | L | D | P | D | A | M | Y | L | D | A | Q | A | Y | E | H | N | Q |

**Notes:**
Name of SARS-CoV-2 proteins (bold) are followed by related genes (italic) and amino acid variations are colour-shaded.
Amino acid variations are shown starting from translated region of each protein.

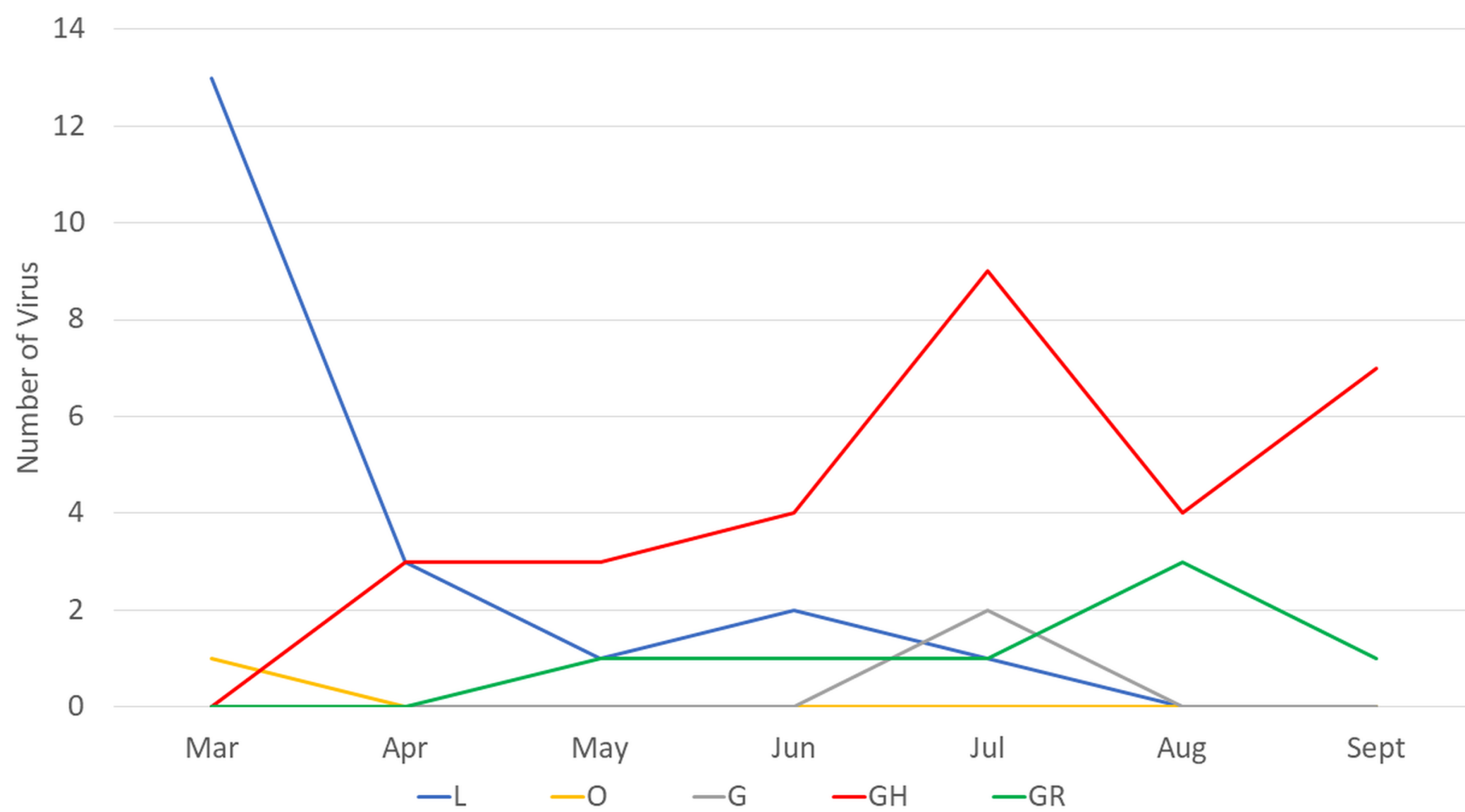

**Figure 1** **Clade distribution of SARS-CoV-2 genomes in Indonesia until the submission date of September 2020, showing that 65% contained the D614G mutation. All G clades (G, GH and GR) carried the D614G mutation.**

consisting of 11 genes (ORF1ab, S, ORF3a, E, M, ORF6, ORF7a, ORF7b, ORF8, N, ORF10).

Next, we compared the clade distribution of full-length genome sequences from Indonesia ($n$ = 60) from March to September 2020. Based on the collection data, most (39/60, 65%) virus genomes contained the D614G mutation representing clade G (2), GR (7) and GH (30) (Fig. 1). From March to April 2020, clade L was dominant. On the other hand, there has been an increase in the detection of clade GH since April 2020 until now.

### Phylogenetic analysis

Phylogenetic analysis of whole genome sequencing showed that three virus samples (EPI_ISL_525492, EPI_ISL_516800 and EPI_ISL_516829) clustered amongst viruses from the clade GH from multiple countries across Asia, Middle East and Europe (Fig. 2). In particular to EPI_ISL_516829, this virus showing an extended branch length to the ancestral human SARS-CoV-2 virus, hCoV-19/Wuhan/Hu-1/2019. On the other hand, one virus sample (EPI_ISL_516806) was situated between clade L viruses mainly from Asia (China, Malaysia, Indonesia, India, United Arab Emirates and Japan) (Fig. 2)

## DISCUSSION

The present study reports four full genomes of SARS-CoV-2 from patients with COVID-19 in Yogyakarta and Central Java Provinces, Indonesia. Phylogenetic analysis showed that three of four samples collected in June and August 2020 were clustered within SARS-CoV-2 viruses belonging to GH clade. One of these (EPI_ISL_516829) displayed a longer branch length compared to the other viruses indicating a greater evolutionary distance to the ancestral virus, hCoV-19/Wuhan/Hu-1/2019. This was confirmed by the

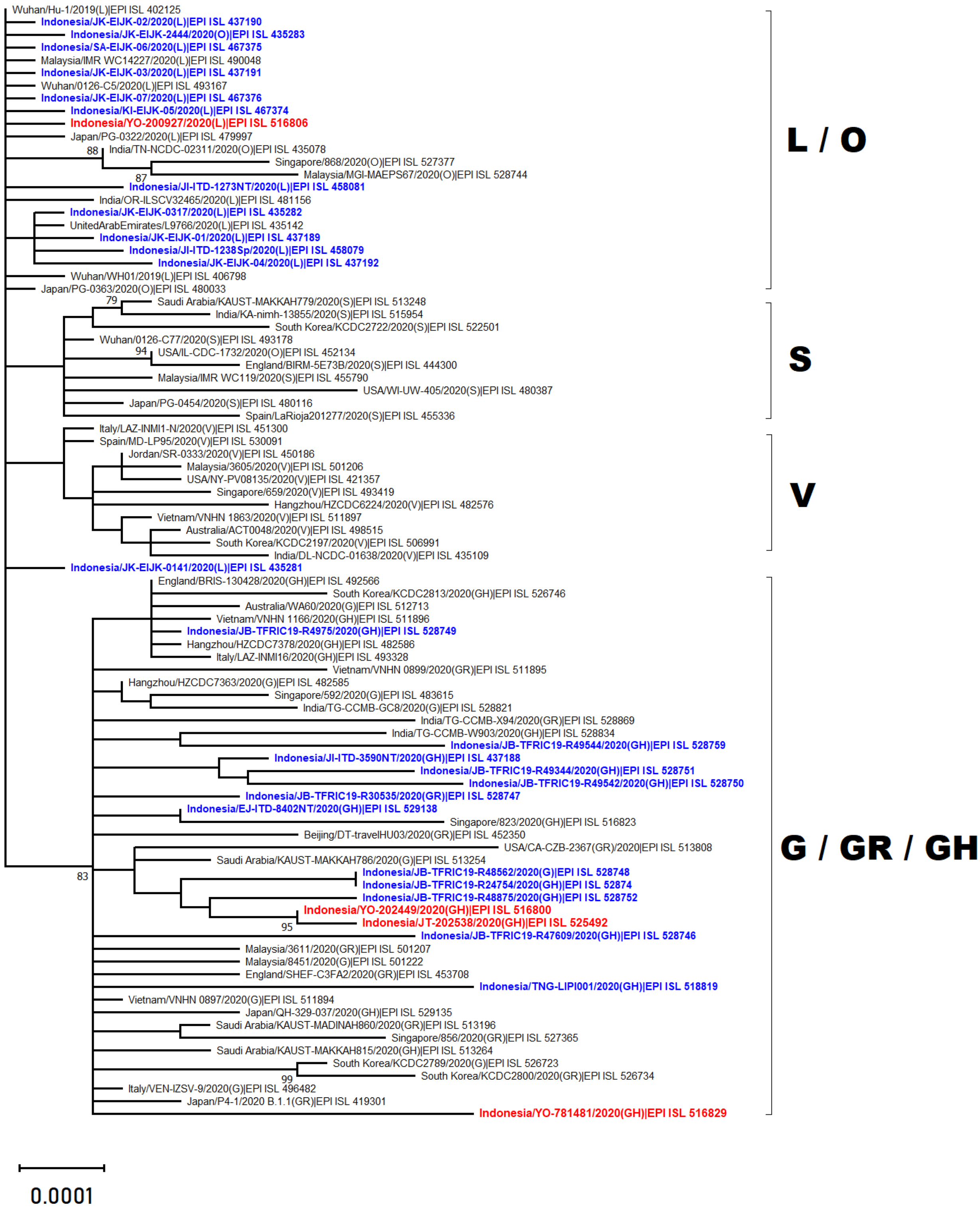

**Figure 2 Phylogenetic analysis of SARS-CoV-2 genomes from Indonesia and different countries.** A phylogenetic tree was constructed from 29.400 nt length of the open reading frame (ORF) of SARS-CoV-2 using the maximum likelihood statistical method with 1,000 bootstrap replications and best-fitting DNA substitution model (GTR+G+I) for the dataset. Virus samples from Yogyakarta and Central Java are indicated in red, while those from other regions in Indonesia are indicated in blue. The tree is drawn to scale (0.0001), with branch lengths measured in the number of substitutions per site.

molecular characterization showing that more SNPs and amino acid mutations were found within this virus genome. The phylogenetic analysis also revealed that one virus sample collected in May 2020 (EPI_ISL_516806) was in evolution closely related to the other SARS-CoV-2 viruses belonged to the L clade, including those from Indonesia which were collected in the early months of disease transmission during the pandemic between March and April 2020.

Our finding corresponds with the situation in Indonesia, showing that during the early pandemic only two clades, O and L, were detected, and the latter clade was more dominantly found from COVID-19 cases. However, since the first detection of clade GH in April 2020, this virus was more frequently detected than the previous circulating clades. Based on the data available in GISAID, 60 virus samples representing five clades have been detected from COVID-19 cases in Indonesia up to September 2020 (based on full-length genome and submission date): L (20), O (1), G (2), GR (7) and GH (30) (*GISAID, 2020*). Whether this pattern correlates with the increase in the number of COVID-19 cases recently in Indonesia has to be investigated further. Interestingly, a similar situation was found in some countries in North America (*Mercatelli & Giorgi, 2020*) and Africa (*Wruck & Adjaye, 2020*), which also detected more SARS-CoV-2 virus strains belonging to clade GH than to the other clades. An increase in SARS-CoV-2 detection conveys the D614G mutation concurrent with the recent global situation of COVID-19 (*GISAID, 2020*).

The D614G mutation dominates globally approximately 77,818/96,215 (~81%) full genomes submitted at GISAID until 18 September 2020 (*GISAID, 2020*). Three of four (75%) SARS-CoV-2 in our case series also consisted of D614G. According to phylogenetic tree and sequence distribution analysis, it has been suggested that the dominating D614G globally is caused by a positive selection (*Korber et al., 2020*), while the dominating D614G in Europe is due to a founder effect (*Dearlove et al., 2020*). Whether which mechanism occurs in Indonesia is difficult to conclude since only limited full genomes were submitted to GISAID until the submission date of the end of September 2020 ($n$ = 60) (*GISAID, 2020*). The virus with the D614G mutation in Indonesia was first detected in April 2020 in Surabaya, East Java (*GISAID, 2020*), followed by other provinces, including Yogyakarta, Central Java, West Java and Banten. Clade L was mostly detected in Jakarta (7/20) and Surabaya, East Java (7/20), followed by Papua (3/20) (Fig. 1) (*GISAID, 2020*).

It has been reported that COVID-19 patients with the D614G mutation have a higher viral load than patients infected by SARS-CoV-2 without mutations (*Korber et al., 2020*). The patients with D614G had a $C_T$ value lower than one patient without the mutation (Table 1).

Interestingly, patients infected with SARS-CoV-2 bearing D614G mutations showed moderate COVID-19, while the patient without mutations suffered from mild symptoms. These differences might be associated with the small sample size of our study ($n$ = 4) compared with previous studies ($n$ = 999 (*Korber et al., 2020*), 175 (*Wagner et al., 2020*) and 88 (*Lorenzo-Redondo et al., 2020*)). Moreover, the severity of COVID-19 is affected by many factors, including age, sex, presence of comorbidities, and patients' immune

responses (*Beeching, Fletcher & Fowler, 2019*; *Zou et al., 2020*). Further study with a larger sample size and involving risk factors for COVID-19 severity is mandatory to determine the association between the D614G mutation and the severity of COVID-19, particularly in Indonesia.

Among GH clades, they also consisted of different mutations in addition to the variants that determine the clade name (Table 1). It has already been reported that the D614G variant is almost always accompanied by three other variants: a C–T change in the 5'UTR, a silent c.3307C > T variant, and P323L (*Younes et al., 2020*). All GH clade samples in the present study also contained P323L (Table 1).

Notably, whole genome sequencing is of practical importance to determine virus variants and clades and is associated with particular geographic disseminations to decide clinical and political approaches at the regional and local levels (*Mercatelli & Giorgi, 2020*). Moreover, whether the differences in the case fatality rate and viral spread or transmission among different countries/regions are affected by differences in the virus clade (*Brufsky, 2020*) needs to be further studied.

Our study has some limitations, including that the samples sequenced are only 0.02% (60/290,000) of all confirmed cases in Indonesia. These facts should be considered in the interpretations of our findings, especially about epidemiological patterns (e.g., increase vs. decrease of frequency) of a particular clade in Indonesia. Another limitation of our study is we do not have any data from epidemiological tracing, nor data concerning detection (how far, how fast) to determine what is the size of the cluster where each patient belongs, or what is the transmission pattern of the cluster.

## CONCLUSIONS

We report the full-genome sequence characterization and phylogenetic analysis of SARS-CoV-2 from Indonesia. SARS-CoV-2 with the D614G mutation appears to become the major circulating virus in Indonesia, which is concurrent with the COVID-19 situation worldwide. Further study with a larger sample size is necessary to investigate whether the dominating SARS-CoV-2 bearing the D614G mutation is due to a positive selection or a founder effect or some other mechanism and to explore the role of the D614G mutation in the pathogenesis and virulence of SARS-CoV-2.

## ACKNOWLEDGEMENTS

We thank the Collaborator Members of the Yogyakarta-Central Java COVID-19 study group: Kurniyanto (RSUP Dr. Soeradji Tirtonegoro), Indah Juliana (RSUP Dr Soeradji Tirtonegoro), Beby Dewi Sartika (RSUD Nyi Ageng Serang), Ardorisye Saptaty Fornia (RS PKU Gamping), Dwiki Afandy (Faculty of Medicine, Public Health and Nursing, Universitas Gadjah Mada (FK-KMK UGM)), Susan Simanjaya (FK-KMK UGM), William Widitjiarso (FK-KMK UGM), Aditya Rifqi Fauzi (FK-KMK UGM), Safitriani (PT. Pandu Biosains), Muhammad Taufiq Soekarno (PT. Pandu Biosains) and Sri Fatmawati (FK-KMK UGM). Gunadi, Marcellus, Dwi Aris Agung Nugrahaningsih, Kristy Iskandar, Nungki Anggorowati, Alvin Santoso Kalim, Kemala Athollah, Dyah Ayu Puspitarani are members of the Genetics Working Group (Pokja Genetik), Faculty of

Medicine, Public Health and Nursing, Universitas Gadjah Mada. We gratefully acknowledge the authors, the originating and submitting laboratories for their sequence and metadata shared through GISAID.

### Funding

This study was funded by the Indonesian Ministry of Research and Technology/National Agency for Research and Innovation (World Class Research 2020 scheme: 3850/UN1/DITLIT/DIT-LIT/PT/2020). The funders had no role in study design, data collection and analysis, decision to publish, or preparation of the manuscript.

### Grant Disclosures

The following grant information was disclosed by the authors:
Indonesian Ministry of Research and Technology/National Agency for Research and Innovation (World Class Research 2020 scheme): 3850/UN1/DITLIT/DIT-LIT/PT/2020.

### Competing Interests

The authors declare that they have no competing interests.

### Author Contributions

- Gunadi conceived and designed the experiments, analyzed the data, prepared figures and/or tables, authored or reviewed drafts of the paper, and approved the final draft.
- Hendra Wibawa conceived and designed the experiments, analyzed the data, prepared figures and/or tables, authored or reviewed drafts of the paper, and approved the final draft.
- Marcellus performed the experiments, analyzed the data, prepared figures and/or tables, and approved the final draft.
- Mohamad Saifudin Hakim performed the experiments, authored or reviewed drafts of the paper, and approved the final draft.
- Edwin Widyanto Daniwijaya performed the experiments, authored or reviewed drafts of the paper, and approved the final draft.
- Ludhang Pradipta Rizki performed the experiments, authored or reviewed drafts of the paper, and approved the final draft.
- Endah Supriyati performed the experiments, analyzed the data, authored or reviewed drafts of the paper, and approved the final draft.
- Dwi Aris Agung Nugrahaningsih conceived and designed the experiments, authored or reviewed drafts of the paper, and approved the final draft.
- Afiahayati performed the experiments, analyzed the data, prepared figures and/or tables, authored or reviewed drafts of the paper, and approved the final draft.
- Siswanto performed the experiments, analyzed the data, authored or reviewed drafts of the paper, and approved the final draft.
- Kristy Iskandar analyzed the data, authored or reviewed drafts of the paper, and approved the final draft.

- Nungki Anggorowati analyzed the data, authored or reviewed drafts of the paper, and approved the final draft.
- Alvin Santoso Kalim performed the experiments, authored or reviewed drafts of the paper, and approved the final draft.
- Dyah Ayu Puspitarani performed the experiments, authored or reviewed drafts of the paper, and approved the final draft.
- Kemala Athollah performed the experiments, authored or reviewed drafts of the paper, and approved the final draft.
- Eggi Arguni performed the experiments, authored or reviewed drafts of the paper, and approved the final draft.
- Titik Nuryastuti conceived and designed the experiments, performed the experiments, authored or reviewed drafts of the paper, and approved the final draft.
- Tri Wibawa conceived and designed the experiments, authored or reviewed drafts of the paper, and approved the final draft.

## Human Ethics

The following information was supplied relating to ethical approvals (i.e., approving body and any reference numbers):

The Medical and Health Research Ethics Committee of the Faculty of Medicine, Public Health and Nursing, Universitas Gadjah Mada/Dr. Sardjito Hospital approved this study (KE/FK/0563/EC/2020). Written informed consent was obtained from all participants before joining in this study.

## DNA Deposition

The following information was supplied regarding the deposition of DNA sequences:

The SARS-CoV-2 genome sequences used in this study are available via GISAID (https://www.epicov.org/epi3/frontend#142ddf):

EPI_ISL_498515, EPI_ISL_512713, EPI_ISL_452350, EPI_ISL_444300, EPI_ISL_492566, EPI_ISL_453708, EPI_ISL_482576, EPI_ISL_482585, EPI_ISL_482586, EPI_ISL_435109, EPI_ISL_515954, EPI_ISL_481156, EPI_ISL_528821, EPI_ISL_528834, EPI_ISL_528869, EPI_ISL_435078, EPI_ISL_529138, EPI_ISL_528745, EPI_ISL_528747, EPI_ISL_528746, EPI_ISL_528748, EPI_ISL_528752, EPI_ISL_528751, EPI_ISL_528750, EPI_ISL_528759, EPI_ISL_528749, EPI_ISL_458079, EPI_ISL_458081, EPI_ISL_437188, EPI_ISL_437189, EPI_ISL_435281, EPI_ISL_437190, EPI_ISL_437191, EPI_ISL_435282, EPI_ISL_437192, EPI_ISL_467376, EPI_ISL_435283, EPI_ISL_525492, EPI_ISL_467374, EPI_ISL_467375, EPI_ISL_518819, EPI_ISL_516806, EPI_ISL_516800, EPI_ISL_516829, EPI_ISL_493328, EPI_ISL_451300, EPI_ISL_496482, EPI_ISL_419301, EPI_ISL_479997, EPI_ISL_480033, EPI_ISL_480116, EPI_ISL_529135, EPI_ISL_450186, EPI_ISL_501206, EPI_ISL_501207, EPI_ISL_501222, EPI_ISL_455790, EPI_ISL_490048, EPI_ISL_528744, EPI_ISL_513196, EPI_ISL_513248, EPI_ISL_513254, EPI_ISL_513264, EPI_ISL_483615, EPI_ISL_493419, EPI_ISL_516823, EPI_ISL_527365, EPI_ISL_527377, EPI_ISL_506991, EPI_ISL_522501, EPI_ISL_526723, EPI_ISL_526734, EPI_ISL_526746, EPI_ISL_455336, EPI_ISL_530091, EPI_ISL_435142, EPI_ISL_513808, EPI_ISL_452134, EPI_ISL_421357, EPI_ISL_480387,

EPI_ISL_511894, EPI_ISL_511895, EPI_ISL_511896, EPI_ISL_511897, EPI_ISL_493167, EPI_ISL_493178, EPI_ISL_402125, EPI_ISL_406798.

To access the sequence data GISAID requires user registration (https://www.gisaid.org/registration/register/).

## Data Availability

Sequences are available in the Supplemental Files.

## Supplemental Information

Supplemental information for this article can be found online at http://dx.doi.org/10.7717/peerj.10575#supplemental-information.

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
