# Peer review of "Full-length genome characterization and phylogenetic analysis of SARS-CoV-2 virus strains from Yogyakarta and Central Java, Indonesia"

_PeerJ, doi:10.7717/peerj.10575_

## Round 0.1 · original submission · Major Revisions

We have received 3 reviews - all demanding major revision. I believe this material is important as a case reports for Indonesia. But the materials presentation should be updated. Please rewrite the text taking in to account all the comments. Please update the references, considering recent publications on SARS-CoV2. Welcome resubmitting the manuscript soon.

Reviewer 1 ·

Basic reporting

no comment

Experimental design

no comment

Validity of the findings

no comment

Additional comments

In this paper, entitled "Full-length genome characterization and phylogenetic analysis of SARS-CoV-2 virus strains from Indonesia", Gunadi et al. reported new 4 whole genome sequences of SARS-CoV-2 from patients in Yogyakarta region, Indonesia, in addition to the 60 other full sequences from Indonesia that have already been deposited in GISAID. This work may pioneer the full genome analysis reports of SARS-CoV-2 in Indonesia. While I appreciate this paper, I have a number of important comments/suggestions, and questions to the authors.

1. The lack of details on the bioinformatics analysis related to the read mapping and SNP calling is concerning:
a) In line 219, the authors mentioned the use of "BWA or Bowtie" sequence alignment methods. Can the authors elaborate further? Which program did the authors actually use? Were both used? In which manner? Or was just one algorithm used?
b) The base calling program/algorithm was not written in the Methods. I suggest to add information related to this matter, including the calling parameters and thresholds, and whether all SNPs called satisfied the thresholds.
c) What is the mean coverage of each sequence reported here?

2. The focus of this paper is to report the full genome of SARS-CoV-2 in Yogyakarta and its characterization. However, majority of the Results section is about patients' information (Case 1 - 4) in a great detail, which deviates from the focus itself. I suggest to put the patient's information as Supplementary Materials, and instead, develop more on the genetic analysis results in the Main Text.

3. In Figure 3 (Phylogenetic Tree analysis),
a) what is the length unit of the tree?
b) I suggest rooting the tree, to be able to observe the ancestrality.

4. In line 228, the authors claimed that there is an increase of clade GH detection since April 2020 until now. In the sense that the samples sequenced are only 0.02% of all confirmed cases in Indonesia (60/290000), and with the current rate of 4000 new confirmed cases perday, 60+4 samples (over the range of collection date between March-September) cannot robustly conclude about epidemiological pattern (eg. increase vs. decrease of frequency) of a particular clade in indonesia.

5. In line 264, the authors miscited reference number 10 (Dearlove et al. 2020; PNAS). Dearlove et al. mentioned the founder event in Europe, but not globally (see Dearlove et al.'s Figure 2 caption). Meanwhile the global spread of the D614G mutation is more likely due to positive selection (see Korber et al. 2020; Cell).

6. In line 273, the authors' mentioned the correlation between the severity of the cases vs. the 614G mutation pattern, based ONLY on 4 individuals. Furthermore, severity is caused by multi-factors, including host immunity, treatment, etc. that were not considered here. The authors must clearly mention the caveat, or completely remove the sentence.

7. Reference number 12 (Lorenzo-Redondo et al. 2020) was never cited in the Main Text.

Reviewer 2 ·

Basic reporting

The Introduction and basic support are not enough.

Experimental design

No comment, I have add in the comments to the authors.

Validity of the findings

If the authors analyzed only 4 genomes, The findings is not enough for publication.

Additional comments

The authors reported and described the four full-length genome sequences of SARS-CoV-2 characterization and compared them with the previous genome in Indonesia from the GISAID. The SARS-CoV-2 with D614G mutation has been the significant circulating strain worldwide since March 2020. We now have the whole genome of more than 10,000 genomes worldwide in GISAID. The analysis of the data should be reanalyzed with the other whole genome in Indonesia. The title should be changed. If you analyze only your four genomes; there is not enough scientific significance for publication. I would suggest you reanalyze all of them in Indonesia with more information on location, date collection and etc.

I have some comments about your MS in this study as follows

Introduction

1. The introductions needed to be more detailed. The authors should explain more about the clinical epidemiology of COVID-19 in Indonesia, for example, How was the first case identified, the peak of transmission, diagnostic tests, public health interventions, etc.
2. Please add a reference (s) at the end of the sentence in Line 98 and 100.
3. Spelling mistakes: sydromecoronavirus
4. Your aim to compare with the other whole genome sequence in Indonesia, but you did not do the molecular characterization for them.

Methods
1. Criteria to select these 4 cases to perform NGS?
2.The authors should describe the clinical criteria of mild, moderate and severe cases of COVID-19

Results
1. The reported 4 cases can be summarized in table 1. The results of point mutations with the other whole genome of Indonesia can be described to see the whole picture of Indonesian strains. Table 1. In the manuscript, the explanation for “Lineage B.1.36 and B” in table 1 is missing.

1. Line 180. what is the full name of NLR.
- (Page 9, line 112). Give manufacturer, city, country for VTM and all the products or reagents should be company, city, country except the USA use the only abbreviation of the state.
- (Page 10, line 141). Why not maximum likelihood?

2. Fig 1 does not provide additional information and justify why you want to show the photos of a circular map of four SARS-CoV-2 genomes; they do not show anything new about the genome, except maybe the presence of the original genome, but even that has already been published for other papers. The authors should consider removing it
2. Clade distribution (Fig 2). The authors should define the unit on the y-axis. Is it percent or number of persons?

Discussion
1.Line 274-276, the authors should discuss in more detail about the mutations of SARS-CoV-2 and clinical severity, especially the part when the authors compare this study's sample size with other studies.
2.(Page 13, line 254-256). If you are making a statement about the increase of GH clade in North America and Africa, please add at least one reference from America or Africa.
3.This paper lacks a statement of the major limitations of the findings before the concluding paragraph.

·

Basic reporting

The study basically reported 4 SARS-CoV-2 genome sequences originating from Yogyakarta, Indonesia. The authors also analyzed global as well as other Indonesian sequence entries.

English is relatively understandable with a few require more clarity and grammatical corrections:
1. What is NLR: Neutrophil-to-Lymphocyte Ratio? Please also check for other unexplained abbreviations.
2. Line 98 in Introduction mentioned an vague expression "major frequency"? Does this mean "most frequently detected"? This statement needs to be more substantiated. How many percent?
3. Line 204-205 should read "He experienced..."

Background is succinct and sufficient. Raw data was not needed because specific accession IDs were given to the database.

Suggestions on figures/tables:
1. Figure 2 is rather confusing. Please explain that D614G mutation is carried by all G clades (G, GH and GR). Does the Y-axis represent percentage or number of genomes?
2. In Table 2, does ID refer to GISAID Accession ID?

Experimental design

The manuscript reported original research within the aims and scope of the journal. Research question as reflected in the objectives should focus on viral genome characteristics in Indonesia, both among different areas and with other countries. There are additional requirements to investigation:
Ethical standard was fulfilled.

There are concerns with regards to methodology:

1. It was not explained why only these 88 genomes were selected, while presumably sequences of all genomes were available. Were the Indonesian sequences all that were submitted as per the retrieval date? My current quick search with GISAID gave 111 sequence entries from Indonesia. I think it is only reasonable to conclude on Indonesian mutation situation based on analyses of all sequences submitted from the country. Besides, I am not sure how much sequences from global entries will add value to this study, except all were analyzed. If analyzing global sequence entries is not possible, perhaps better to focus on providing and analyzing (all) Indonesian sequence data.

2. The biggest concern regarding D614G is that the mutation leads to quicker transmission. I think having more detail patient information leading to this information (especially those reported by the manuscript, as specified in Table 1) will add value to the manuscript: epidemiological tracing data upon detection (how far, how fast), what is the size of the cluster where the patient belongs, how is the transmission pattern of the cluster.

Validity of the findings

I find it difficult to comment on this because of the first concern in methodology. There was unfocused data for analyses. Why these 88? Why only selected Indonesian entries?

Additional comments

I commend efforts by the authors' group for their success in obtaining the viral sequence, analyzing and reporting the data. Few more works are required before the manuscript can be published.

---

## Round 0.2 · Minor Revisions

Please update the text according to the comments by Reviewer #1. This doesn't need a new reviewing round. Just take into account all the comments in the final version.

As the academic editor I’d recommend adding recent references on publications on COVID-19.

Thanks again for discussing this important problem.

Reviewer 1 ·

Basic reporting

no comment

Experimental design

no comment

Validity of the findings

no comment

Additional comments

I thank the authors for addressing my comments in previous submission. However there are a couple of (minor) points that still need to be clarified:

1. I'm not familiar with UGENE, but the variant calling parameters and thresholds are still not specified in this revised manuscript. However considering the average depth is high, I am assuming the variants called are correct.

2. The tree unit is still not described. What does "0.0001" mean in the tree scale? What is the unit of that scale?

3. I noticed inconsistencies in the current Figure 1 (previously Figure 2): the "clade distribution of SARS-CoV-2 genomes in Indonesia ...".
a) the shapes are ambiguous when overlapped, they become indistinguishable. I think It's better to have the graph in color rather than in shapes for this particular case.
b) many numbers are inconsistent between the two figures.
b1) in prev Figure, in August, clade GH is at 3, while in the revised Figure, it is at 4. The inconsistency persists in clade GH in September.
b2) clade GR distribution is *very* different between the two figures.
b3) similarly, clade L distribution is also inconsistent

·

Basic reporting

Ok. No further comments

Experimental design

Ok. No further comments

Validity of the findings

Limitations were mentioned. No further comments

Additional comments

No further comment

---

## Round 0.3 · accepted · Accept

Thanks for the update. I have no more remarks on the manuscript.